# Nutritional and Nutrition-Related Biomarkers as Prognostic Factors of Sarcopenia, and Their Role in Disease Progression

**DOI:** 10.3390/diseases10030042

**Published:** 2022-07-06

**Authors:** Sousana K. Papadopoulou, Gavriela Voulgaridou, Foivi S. Kondyli, Mariella Drakaki, Kyriaki Sianidou, Rozalia Andrianopoulou, Nikolaos Rodopaios, Agathi Pritsa

**Affiliations:** 1Department of Nutritional Sciences and Dietetics, School of Health Sciences, International Hellenic University, 57400 Thessaloniki, Greece; gabivoulg@gmail.com (G.V.); fiviks@gmail.com (F.S.K.); drakakimariella@gmail.com (M.D.); agpritsa@ihu.gr (A.P.); 2School of Biology, Aristotle University of Thessaloniki, 54636 Thessaloniki, Greece; kiki_sianidou_95@yahoo.gr; 3Department of Medical Laboratories, School of Health Sciences, University of Thessaly, 41110 Larissa, Greece; rozaliaandrianopoulou@gmail.com; 4Department of Social Medicine, Preventive Medicine and Nutrition Clinic, School of Medicine, University of Crete, Voutes, 71003 Iraklion, Greece; nikow1966@yahoo.gr

**Keywords:** biomarkers, nutrition, sarcopenia, malnutrition

## Abstract

Due to the multifactorial pathogenesis of sarcopenia, it is crucial to identify biomarkers that are risk factors for sarcopenia, and which therefore have a prognostic function. **Aim**: This narrative review aims to define a set of biomarkers associated with nutrition and sarcopenia. These biomarkers could contribute to individualized monitoring and enable preventive and therapeutic methods. **Methods**: Two electronic databases, PubMed and Google Scholar, were used. The search strategy was based on a controlled vocabulary (MeSH) and includes studies published up to February 2022. **Discussion**: Higher levels of serum uric acid are associated with higher handgrip strength and better muscle function in elderly people and, thus, may slow the progression of sarcopenia. Leptin, an adipokine secreted by adipose tissue, promotes the production of pro-inflammatory cytokines, which in turn lead to sarcopenia. This makes leptin a significant indirect biomarker for physical disability and sarcopenic obesity. Additionally, creatinine is a reliable biomarker for muscle mass status because of its easy accessibility and cost-effectiveness. Vitamin D status acts as a useful biomarker for predicting total mortality, hip fractures, early death, and the development of sarcopenia. Therefore, there is an increasing interest in dietary antioxidants and their effects on age-related losses of muscle mass and function. On the other hand, 3-Methylhistidine is a valuable biomarker for detecting increased muscle catabolism, as it is excreted through urine during muscle degradation. In addition, IGF-1, whose concentration in plasma is stimulated by food intake, is associated with the loss of skeletal muscle mass, which probably plays a crucial role in the progression of sarcopenia. **Conclusions**: Many nutritional biomarkers were found to be associated with sarcopenia, and can therefore be used as prognostic indexes and risk factors. Nutrition plays an important role in the prevention and management of sarcopenia, affecting muscle mass, strength, and function in elderly people.

## 1. Introduction

Sarcopenia is primarily a geriatric syndrome, which is known for progressively decreasing skeletal muscle mass and function [1]. It is one of the diseases that the elderly are most frequently affected by, and it leads to an increased risk of frailty, falls and injuries that result in hospitalization, a loss of independence, and increased morbidity and mortality [2]. Lean muscle mass is reduced to up to 25% of total body weight by the ages of 75–80 years; as a result of this age-related muscle mass loss, the basal metabolic rate is reduced by 30% in those aged 70 [3,4]. Although sarcopenia develops as a result of several factors (e.g., criteria, measurements), its prevalence in community-dwelling individuals was estimated to be approximately 11% in males and 9% in females; meanwhile, in hospitalized individuals, it was estimated to have a prevalence of approximately 23% in men and 24% in women [5]. 

In 2019, the European Working Group on Sarcopenia in Older People (EWGSOP) defined sarcopenia as the co-existence of low muscle strength, low muscle quality/ quantity, and/or low physical performance [6]. The working group also proposed techniques and methods for the measurement of these parameters, suggesting that muscle strength could be measured by isokinetic or isometric dynamometry [7], and muscle quality or quantity could be estimated using several methods. Although computed tomography (CT) and magnetic resonance imaging (MRI) are the gold standard, the methods of dual-energy absorptiometry (DXA) and bioelectrical impedance analysis (BIA) are more commonly used in clinical practice to measure muscle mass. Finally, physical performance can be measured by gait speed, a timed “up and go” test (TUG), a 400 m walking test, and a short physical performance battery (SPPB) [6].

Common risk factors for sarcopenia include third age [8,9], low levels of physical activity, malnutrition [5], body mass index [9], and several co-morbidities, such as cardiovascular diseases, diabetes, respiratory diseases [10], human immunodeficiency virus (HIV) [11], and cancer [12]. Moreover, multiple pathogenesis mechanisms such as endocrine dysfunction, growth factors, neuromuscular junction, muscle protein changes, and inflammatory conditions increase the complexity of sarcopenia [13]. However, even in light of the important overlaps between the sarcopenia phenotype and comorbidities, the disease is often underdiagnosed in clinical practice [14].

Until now, efforts to prevent and treat sarcopenia have mainly focused on physical activity and nutritional intervention/ therapy. Vitamin D, whey protein, leucine [15,16], and adequate protein intake (1 g/kg/day) [17,18] are among the nutritional factors that have been shown to be most effective in helping sarcopenic patients to maintain or increase their muscle mass and muscle function. However, there is limited information on nutritional element deficiencies in patients with sarcopenia, and to what extent these elements improve nutritional status by preventing sarcopenia [19]. Furthermore, appetite is reduced while aging (“anorexia of ageing”), increasing the risk of malnutrition and sarcopenia [20]. Other studies have demonstrated that resistance training combined with nutritional supplementation could improve muscle quality and muscle strength [21,22,23]. Furthermore, there are currently no medical treatments that are clinically effective in maintaining or increasing muscle mass and/or function in sarcopenic patients.

As a consequence, over the last few years, many reported variables have been identified as potential biomarkers in the evaluation of sarcopenia [24]. Specific blood- and serum-based biomarkers associated with clinical assessments of sarcopenia have been recommended as a tool for detecting and monitoring older adults who are at risk, and for evaluating the effectiveness of strategies that aim to prevent and treat sarcopenia. This review offers an overview of the biomarkers related to nutrition, and aims to contribute to the description of various mechanisms of sarcopenia in different patients. In turn, this will aid the individualized monitoring of the success of preventive and therapeutic methods. 

## 2. Methods

Two electronic databases, PubMed and Google Scholar, were searched to identify relevant articles. For the search strategy, we used the terms “nutritional”, “biomarkers”, “sarcopenia”, “vitamin D”, “n-3 omega fatty acids”, “antioxidants”, “3-Methylhistidine”, “carnintine”, “CRP”, “UTNT”, “leptin”, “carnitine”, “IFG-1”, “visceral proteins”, “albumin”, “pre-albumin”, and “transthyretin”, using the Boolean operator “AND”. Only the articles most relevant to our review, which were published in English up to February 2022, were included. Studies in experimental models and in vitro were excluded. 

We classified biomarkers based on their concentration in blood or serum. Specifically, they were classified into biomarkers whose either increased or decreased concentration in serum or plasma is associated with sarcopenia. 

## 3. Biomarkers with Increased Concentration in Sarcopenia

As mentioned above, elderly people have low appetites, which contributes to adverse outcomes in their nutritional status. Malnutrition is considered a key factor in the development of sarcopenia. Due to the aforementioned factors, the diagnosis of sarcopenia can be supported by many biomarkers. Table 1 shows the relationship of those biomarkers with sarcopenia and nutrition.

### 3.1. Uric Acid

Muscle loss resulting from physical inactivity and sarcopenia in elderly people is affected by oxidative stress [58]. Elevated uric acid levels in the blood are positively associated with the intake of red meat, seafood, alcohol, or fructose, and negatively correlated with dairy or soy products [50]. Uric acid acts as an antioxidant, meaning that increased levels of uric acid could potentially protect against the uncontrolled production of free radicals [59]. Rises in uric acid levels in serum have been shown to increase handgrip strength and lead to greater muscle function in elderly people [51,52,53], which could perhaps decelerate the progression of sarcopenia.

### 3.2. Carnitine

Carnitine (β-hydroxy-N-trimethyl-aminobutyric acid) is an amino acid derivative involved in mitochondrial energy production in skeletal and cardiac muscles. Carnitine is mainly derived from dietary sources, but it is also synthesized from lysine and methionine in the liver, kidneys, and brain. In the metabolic pathway of carnitine in the muscles, free L-carnitine (FC) and acylcarnitine (AC) produce energy through beta-oxidation of fatty acids [31]. An increase of the levels of certain types of acylcarnitine, has been found to be associated with dysregulation of fatty acid metabolism [60]. Dysfunction of metabolic pathways in mitochondria has been reported to contribute to the decrease of muscle mass and loss of muscle strength [61]. In addition, recent studies reveal that certain types of carnitine in blood, are found in insufficient levels in sarcopenic patients and were significantly correlated with the skeletal muscle index (SMI). These findings indicate that carnitine is a potential biomarker for sarcopenia [31,32].

### 3.3. C-Reactive Protein

C-reactive protein (CRP) is an acute-phase protein and a marker of systemic inflammation [36,62]. Ιt has been reported that high levels of CRP can lead to a decrease in the size of muscle cells and downregulate protein kinase B (Akt), which is theoretically related to dysregulation of muscle cells [36,63]. Several studies have shown that increasing levels of inflammatory markers, including CRP, in older adults are associated with increased muscle loss and sarcopenia [37]. Also, high levels of CRP have been reported to be inversely correlated to muscle strength [36,38].

Adipose tissue plays an important role in inflammatory processes because adipocytes are a source of pro-inflammatory cytokines, including IL-6, which affects CRP production. Moreover, CRP levels have been reported to be higher in obese patients than in normal-weight patients. Thus, it seems that obesity contributes to the formation of a pro-inflammatory environment [62]. In addition, several studies recommend that a diet rich in vegetables, fruits, whole grains and nuts has a beneficial effect on reducing CRP levels [64,65].

### 3.4. Urinary Levels of Titin-N Fragment

Titin makes up a significant part of the sarcomere’s structure and is considered to be an indicator of muscle damage, as it is broken down into fragments by proteolytic enzymes and released into the urine in cases of muscle deterioration [66]. Therefore, a larger concentration of titin fragments is found in the urine of people suffering from sarcopenia, making titin a useful biomarker for the condition [67]. In addition, studies have highlighted the importance of titin as a potential biomarker for an individual’s nutritional condition. Specifically, Miyoshi et al., 2020 [54], revealed a negative correlation between urinary titin fragment levels and a variety of nutritional status indicators (Alb, pre-Alb, Cholinesterase, Subcutaneous Fat Volume). Additionally, a negative correlation was found between the protein and skeletal muscle mass; meanwhile, the only positive correlation found concerned aging.

### 3.5. Leptin

Adipose tissue secretes leptin, a prevalent adipokin. Leptin is a 16-kDa protein, the production of which increases during over-nutrition; it is produced by the obesity gene that regulates appetite, insulin sensitivity, inflammation, and fat deposition [42,68]. Age-related increases in fat mass lead to greater concentrations of leptin in the body, which may in turn result in leptin resistance and fatty acid oxidation in muscles. Ectopic fat deposition in organs can result from this, affecting the heart, liver, and muscles, and thereby reducing muscle quality in obese older people [42]. In addition, in the presence of leptin, there is a greater production of pro-inflammatory cytokines, such as tumor necrosis factor-α (TNF-α), interleukin-6 (IL-6) and interleukin-12 (IL-12) [43]. Sarcopenia seems to be linked to these cytokines. Pro-inflammatory cytokines, insulin resistance, oxidative stress, hormonal changes, and functional impairment are pathophysiological mechanisms that sarcopenia and obesity seem to have in common [69]. Cytokines and adipokines promote inflammation which enhances the likelihood of developing sarcopenic obesity [70]. At the same time, as mentioned above, leptin and pro-inflammatory cytokines are secreted by adipose tissue and catabolic processes within the muscles [71]. As a consequence, a dangerous cycle is initiated, which accelerates the development of sarcopenia and, in time, leads to physical disability [70]. 

As indicated previously, there is an association between serum leptin and sarcopenic obesity [51]; the same can be said for elevated levels of CRP, IL-6 and sIL-6r. IL-6 could potentially have inflammatory and anti-inflammatory effects. The distribution of fat within the body is also of great importance, as central obesity promotes inflammation more than general obesity does [49].

### 3.6. Creatinine

Creatinine is the end product of creatine, a three-amino-acid compound primarily detected in muscle [35]. The creatinine excretion rate (CER) can be helpful when it comes to the indirect assessment of body composition in healthy adult subjects [72]. The normal ranges of CER in healthy individuals are 18 to 21 mg/kg Cr in women and from 21 to 25 mg/kg Cr in men [73]. 1.9 kgs of skeletal muscle produce 1 mmol of creatinine in urine [39]. When a protein balance is present, the concentration of creatinine in urine is reduced [74]. Frequent measurements of serum creatinine levels are recommended, as they can indicate the state of the skeletal muscles [13]. The downside of using creatinine as a biomarker is that it is affected by the function of the kidneys, and urine must be collected for 24 h in order for it to be measured [35]. Nonetheless, muscle mass status can be reliably assessed by using creatinine as a biomarker, as it is easily accessible and cost effective [75].

## 4. Biomarkers with Decreased Concentration in Sarcopenia

### 4.1. Vitamin D

Vitamin D is highly important in relation to bone metabolism, muscle strength [76,77] and physical performance [76] as it regulates phosphate and calcium homeostasis. Vitamin D has an impact on the number and diameter of type II muscle cells, especially type IIA. Type IIA cells specifically induce rapid muscle contraction velocity, and are essential for short duration, high-intensity anaerobic activities such as acceleration, deceleration, sprinting, and jumping [56]. Vitamin D levels constitute a helpful biomarker when it comes to the prediction of total mortality, hip fractures, early death, and the development of sarcopenia [57]. However, more research is required to explain in detail how Vitamin D acts in human muscle tissue, and to describe the underlying mechanisms.

### 4.2. n-3 Fatty Acids

N-3 Fatty Acids have anti-inflammatory capacities, and thus protect against chronic metabolic diseases [78]. Muscle loss can be intensified by chronic inflammation, through age-related pro-inflammatory cytokines (such as TNF-α and IL-6) and immune dysfunction [79]. Consequently, the role n-3 FAs play in reducing inflammation can explain how they protect against sarcopenia [45]. Additionally, n-3 FAs can stimulate mTOR signaling, leading to anabolic effects in the skeletal muscles and to muscle protein synthesis [80]. Consequently, n-3 FAs could aid in treating age-related anabolic resistance by increasing the rate of muscle protein synthesis by stimulating the mTOR signaling pathway [80]. In summary, n-3 FAs can protect human muscle homeostasis while reduced n-3 serum levels can lead to an increased risk for sarcopenia [45].

### 4.3. Antioxidants

Research suggests that decreases in muscle strength and sarcopenia during the ageing process are related to oxidative stress. Proteins, fat tissue, and DNA can be damaged by reactive oxygen species (ROS). Antioxidant defense mechanisms, such as glutathione peroxidase and thioredoxin, and exogenous antioxidants that derive from diet, including carotenoids, selenium, flavonoids, tocopherols, and other plant polyphenols, can often reverse the processes of ROS [33]. As a consequence, many researchers are paying attention to dietary antioxidants and their influence on age-related decrease of muscle mass and function.

Low selenium levels also seem to be related to low muscle mass in the elderly [49]. Furthermore, increased carotenoid levels in plasma are associated with a higher risk of increased IL-6 levels [33], and with a lower risk of severe kinetic disability among the elderly [34]. Additionally, certain muscle adaptations to strength training in older men can be undermined by vitamin C and E deficiency [55]. More research is required on the influence of antioxidants on muscle mass and function.

### 4.4. 3-Methylhistidine

The amino acid 3-Methylhistidine (3-MH) [25] is produced by the post-translational methylation of specific histidine residues in actin and myosin—proteins that comprise the myofibrils [81]. It has been estimated that over 90% of 3-MH in the body is found in skeletal muscle [25]. When these myofibrillar proteins undergo proteolysis, 3-MH is released into the circulation and is then excreted in urine without being metabolized [25,26]. Muscle protein degradation is the only endogenous source of 3-MH in human plasma, and the excretion of 3-MH in urine has been used to calculate the rate of skeletal muscle degradation [25]. Thus, it is suggested that 3-MH might represent a potential biomarker for muscle protein turnover, which could contribute to the diagnosis of sarcopenia and frailty [26]. However, tissues other than skeletal muscle are able to release 3-MH in urine [25], and the presence of 3-MH in the urine is influenced by the ingestion of medication [82]. In addition, plasma 3-MH might be affected by food intake. Meat, fish, and their products are the only dietary exogenous sources of 3-MH, and can influence plasma concentrations of 3-MH. Therefore, a 24-h meat-free period is recommended before blood sampling, but this may pose a challenge in a clinical context [26].

### 4.5. Visceral Proteins

Malnutrition in the elderly can become evident via serum proteins such as: (a) pre-albumin, (b) albumin, (c) transferrin, and (d) retinol-binding protein (RBP).

The liver produces pre-albumin (or transthyretin, TTR), a thyroid hormone-transport protein, which is later partially catabolized by the kidneys. Serum pre-albumin levels of less than 10 mg/dL are associated with malnutrition [46]. Using pre-albumin as a nutritional biomarker in the elderly, especially while re-feeding, seems to be encouraged. The pre-albumin parameter is positively correlated to the loss of muscle mass in elderly people. The liver produces less pre-albumin both when it is in a protein-deficient state, and in the presence of cytokine-induced inflammatory disorders. Thus, when pre-albumin values are reduced, they can be used to define the severity of involutional lean body mass (LBM) processes, by distinguishing whether the abnormalities that have resulted from protein restriction are hepatic or muscular [47]. Despite the pre-albumin levels being easily affected by nutritional deficiencies, they should still be included during a thorough evaluation [46].

Muscle mass and function seem to decrease when albumin levels in serum are low [27,28,29]. Additionally, these values have been used to detect malnutrition for decades. Hypoalbuminemia is a term used to describe low albumin serum levels, and is used to predict the mortality of elderly people who are either community-dwelling, residents in nursing homes, or hospitalized. Inflammation, specifically that which is characterized by high concentrations of IL-6 and TNF-α, impacts hypoalbuminemia [29]. The concentration of albumin declines when: (a) nephrotic syndrome is present, (b) synthesis declines as a result of inflammatory cytokines or liver failure, or (c) enteropathies and gastrointestinal disorders are present [30]. However, the American Society for Parenteral and Enteral Nutrition suggests that both albumin and pre-albumin should be associated with inflammation rather than malnutrition, as they cannot reflect the nutrition status of a patient [83]. The association between inflammation and malnutrition is the reason that these visceral proteins are mistakenly cited as nutrition biomarkers [83]. As such, serum albumin and pre-albumin should be viewed as inflammatory biomarkers which are used to identify nutrition risks in patients. A patient is at nutrition risk if he or she lacks adequate nutrition support, which could lead to malnutrition and poor outcomes [83].

In contrast, albumin and pre-albumin levels in serum are sustained when healthy individuals are severely deprived of nutrients and experience significant weight loss, either due to limited access to food or an unwillingness to eat, primarily because of anorexia nervosa. Only when the malnutrition is severe (BMI < 11 kg/m^2^), is a decrease in levels observed [84]. In summary, the evidence is inconclusive when it comes to visceral serum proteins and their ability to predict nutritional deprivation.

Transferrin can also serve as an index of nutritional status [85]. Transferrin is considered to play a useful role in assessing nutritional status by some researchers [86], but not by others [87]. When malnutrition is severe, serum transferrin levels decline. However, this marker cannot accurately assess mild malnutrition and lean mass in the elderly [48].

Retinol-binding protein (RBP) is a protein needed for the successful transfer of retinol from the liver to the target organs [88]. Deficiencies in retinol can potentially cause serum levels of RBP to decrease because its production in the liver is reduced [89]. Despite this, patients suffering from nephropathy, who are malnourished in protein and calories, can have levels of serum RBP within the normal range [90]. On the other hand, underweight subjects display mean values of pre-albumin and RBP that are substantially reduced, and are more highly correlated with fat-free mass (FFM), in comparison to albumin [39]. Therefore, when compared to the benefits that pre-albumin measurement has to offer, RBP measurement is not considered a useful tool [35].

### 4.6. Insulin-like Growth Factor (IGF-1)

IGF-1 is mainly produced in the liver [40]. Plasma IGF-1 levels decrease during fasting, while food intake stimulates its concentration [40]. There seems to be an association between energy intake (and, partially, protein intake) and plasma IGF-1 levels [91]. Renal dysfunction, liver disease, and severe trauma alter IGF-1 concentration [39]. Additionally, various anabolic processes within the skeletal muscle are aided by insulin-like growth factor-1 (IGF-1), while low IGF-1 concentrations are related to skeletal muscle atrophy, suggesting that it could possibly be a key component in the development of sarcopenia [41]. The acute phase response, among other factors, can affect IGF-1 serum concentrations, creating a limitation when it comes to measuring it [35].

## 5. Conclusions

Malnutrition, and other factors such as oxidative stress, physical inactivity, and inflammation are some of the risk factors that are associated with the development of sarcopenia. Despite the fact that aging and sarcopenia have common molecular and cellular mechanisms, there is no conclusive pathophysiological field dedicated to these conditions. Since multiple factors can lead to the development of sarcopenia, it is vital that we recognize the importance of identifying different biomarkers. A balanced diet could play a crucial role in preserving muscle mass, strength, and function in older adults. Malnutrition is widespread amongst older patients, highlighting the importance of ensuring adequate dietary intake and satisfactory nutritional status for these populations. Diagnosing and treating sarcopenia in the early stages can be assisted by routine screening and early diagnosis of malnutrition. Biomarkers associated with nutrition (including albumin, vitamin D, and IGF-1) can be used to evaluate nutritional status and muscle mass, strength, and function in individuals, aiding in the identification and treatment of sarcopenia. More studies are required in order to determine other potential biomarkers for sarcopenia.

## Figures and Tables

**Table 1 diseases-10-00042-t001:** Characteristics of biomarkers and their relation to sarcopenia and nutrition.

Biomarker	Type of Molecule/Biological Domain	Association with Nutrition	Association with Sarcopenia	Other Associated Disease	References
3-methylhistidine	Amino acid	Dietary intake	Increased levels associated with muscle protein degradation	-	[25,26]
Albumin	Protein	Associated with malnutrition	Low serum levels lead to a decrease in muscle mass and function	Lower levels associated with nephrotic syndrome, inflammatory cytokines, liver failure, enteropathies, and gastrointestinal disorders	[27,28,29,30]
Carnitine	Amino acid	Mainly dietary intake	Decreased levels associated with sarcopenia and SMI	-	[31,32]
Carotenoids	Hydrocarbons	Dietary intake	Increased levels associated with lower risk of disability in walking	Increased levels associated with higher risk of increased IL-6 level	[33,34]
Creatinine	Breakdown product of creatine	In a protein-balanced diet, the concentration of creatinine in urine is reduced	High concentration in urine associated with muscle degradation	-	[35]
C-reactive protein (CRP)	Protein	Affected by obesity and quality of diet	Increased levels associated with muscle loss, lower muscle strength, and sarcopenia	Systemic inflammation	[36,37,38]
Insulin-like growth factor (IGF-1)	Hormone	Decreased during fasting, affected by food intake	Related to skeletal mass atrophy, possibly a factor of sarcopenia development	Renal dysfunction, liver disease, severe trauma	[39,40,41]
Leptin	Protein hormone	Associated with overnutrition	When present, higher production of pro-inflammatory cytokins linked to sarcopenia and sarcopenic obesity		[42,43,44]
n-3 fatty acids	Fatty acids	Dietary intake	Decreased serum levels associated with lower risk of sarcopenia	Reduced inflammation	[45]
Pre-albumin	Protein	Less than 10 mg/dL associated with malnutrition; affected by protein restrictive diets	Decreased levels associated with reduced muscle mass and LBM	Higher in protein deficient state; higher in cytokine-induced inflammatory disorders	[46,47]
Retinol binding protein (RBP)	Protein	Reduced when underweight	Related to malnutrition		[48]
Selenium	Trace element	Dietary intake	Decreased levels associated with low muscle mass	-	[49]
Transferrin	Protein	Is an index of nutritional status	Related to malnutrition		[48]
Uric acid	Purine derivative	Affected by food intake	Increased levels associated with increased handgrip strength and greater muscle function	-	[50,51,52,53]
Urinary levels of Titin-N fragment	Protein fragment	Negative correlation with nutrional status indicators	Patients with sarcopenia have higher concentration in their urine	-	[54]
Vitamins C and E	Vitamins	Dietary intake	Undermines muscle adaptations to strength training	-	[55]
Vitamin D	Vitamin	Dietary intake	Affects number and diameter of type II muscle cells	Total mortality, hip fractures, and early death	[56,57]

SMI = skeletal muscle index; IL-6 = Interleukin-6; LBM = lean body mass.

## Data Availability

Not applicable.

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
