# Peer review of "Nutritional and Nutrition-Related Biomarkers as Prognostic Factors of Sarcopenia, and Their Role in Disease Progression"

_diseases, 2022, doi:10.3390/diseases10030042_

Round 1
Reviewer 1 Report
Authors intend to define a review of biomarkers associated with nutrition and sarcopenia. According to the Authors, these biomarkers could share individualized monitoring and enable preventive and therapeutic methods. Two electronic databases PubMed and Google Scholar were used.
Table 1, which represents practically the entire review, must be better differentiated: for example, dividing lines can be inserted, the biomarkers can be listed in alphabetical order, the numbers corresponding to the references must be entered.
Some references are reported in the text with the name and not with the number (lines 40, 71, 221)
Reference n. 1 is incomplete
References n.13 and 55 are the same.
The following position paper, relating to visceral proteins, should be questioned: The Use of Visceral Proteins as Nutrition Markers: An ASPEN Position Paper https://doi.org/10.1002/ncp.10588
Reviewer 2 Report
Title: “Nutritional and nutritionally related biomarkers as prognostic factors of sarcopenia and their role in disease progression”.
Version: 27th June 2022.
Journal: Diseases
Reviewer's report:
This is a narrative review aiming to describe several biomarkers relative to nutrition and sarcopenia. The topic of the manuscript is appropriate for the Journal. It could be of interest to investigators and clinicians.. Minor essential revisions are necessary.
Minor essential revisions
Please check the use of abbreviations through the main text, for example
Table 1 SMI, then only in page 5 is explained.
Page 5 Line 145: Che? SFV? What kind of nutritional status indicators are?. Please check this.
Page 7 Line 244 LBM?
Authors mentioned the biomarkers were classificated based in whose increased and other decreased concentration in serum or plasma. However in table 1 are other distribution. Please check the order of biomarkers in the table, and compare with main text. It is needed for better lecture of the paper.
Table is hard to read. A question Why is the reason to choose these references included in the table and no biomarkers specific references of main text.? Any special reason?
Tittle:
The title is accurate and sufficiently descriptive of the content.
The title is consistent with the presented problem and reflects the main message of the study.
Abstract:
The abstract is concise and specific. Purpose of the study is clearly presented. Please check the order of biomarkers presentation through main text.
Introduction:
The background of the study is clear and precise the problem to be solved.
The purpose of the article clearly presented.
Methods:
Sufficient details about the process are provided. Statistical analyses used are appropriate. The methods are appropriate and well described.
Results:
Information is clearly provided.
Discussion:
The authors did a nice job in the discussion providing important gaps in research about nutritional biomarkers of sarcopenia in older people.
The conclusions are logically valid and justified by the evidence presented in the references cited.
References:
There are 31 and all are appropriate and relevant.
References up to date.
Tables and figures:
One table is shown. It has difficulties to read. It no has concordance with main text. Please check this.
Thanks for letting me review this manuscript.
This could be a nice paper.
Level of interest: An article whose findings are important to those with closely related research interests.
Quality of written English: Well.
Statistical review: No.
Declaration of competing interests:
I declare that I have no competing interest.
